# LEARNING KAN-BASED IMPLICIT NEURAL REPRESENTATIONS FOR DEFORMABLE IMAGE REGISTRATION

## ABSTRACT

Deformable image registration (DIR) is a cornerstone of medical image analysis, enabling spatial alignment for tasks like comparative studies and multi-modal fusion. While learning-based methods (e.g., CNNs, transformers) offer fast inference, they often require large training datasets and struggle to match the precision of classical iterative approaches on some organ types and imaging modalities. Implicit neural representations (INRs) have emerged as a promising alternative, parameterizing deformations as continuous mappings from coordinates to displacement vectors. However, this comes at the cost of requiring pairwise optimization, making computational efficiency and seed-dependent learning stability critical factors for these methods. In this work, we propose KAN-IDIR and RandKAN-IDIR, the first integration of Kolmogorov-Arnold Networks (KANs) into deformable image registration with implicit neural representations (INRs). The learnable activation functions of KANs and their inherent suitability for approximating physical systems make them ideal for modeling deformation fields. Our proposed randomized basis sampling strategy reduces the required number of basis functions in KAN while maintaining registration quality, thereby significantly lowering computational costs. We evaluated our approach on three diverse datasets (lung CT, brain MRI, cardiac MRI) and compared it with competing pairwise learning-based approaches, dataset-trained deep learning models, and classical registration approaches. KAN-IDIR and RandKAN-IDIR achieved the highest accuracy among INR-based methods across all evaluated modalities and anatomies, with minimal computational overhead and superior learning stability across multiple random seeds. Additionally, we discovered that our RandKAN-IDIR model with randomized basis sampling slightly outperforms the model with learnable basis function indices, while eliminating its additional training-time complexity. Source code is available in the supplementary material.

## 1   INTRODUCTION

Deformable image registration (DIR) is a fundamental task in the field of medical image processing. Its primary goal is to determine a spatial transformation that aligns one image with another, ensuring that corresponding regions appear in the same locations. This process enables accurate comparison, fusion, or further analysis of imaging data.

Before the advent of deep learning, non-rigid registration relied on iterative optimization techniques. Examples of methods following this approach include Free-Form Deformation (Rueckert et al., 1999), LDDMM (Beg et al., 2005), Demons (Thirion, 1998), Elastix (Klein et al., 2009), ANT (Avants et al., 2011), pTV (Vishnevskiy et al., 2016), and Flash (Zhang & Fletcher, 2019). These approaches are well-established and mathematically grounded, but they require tuning parameters for each image pair and involve substantial computational costs, which makes them less suitable for real-time registration of large images. With recent advances in artificial intelligence technologies, image-to-image architectures such as convolutional neural networks (CNNs) and transformers have become increasingly popular in medical image registration (Balakrishnan et al., 2019a; Chen et al., 2022). These approaches predict displacement vector fields for image pairs, offering inference speeds several orders of magnitude faster than clas-

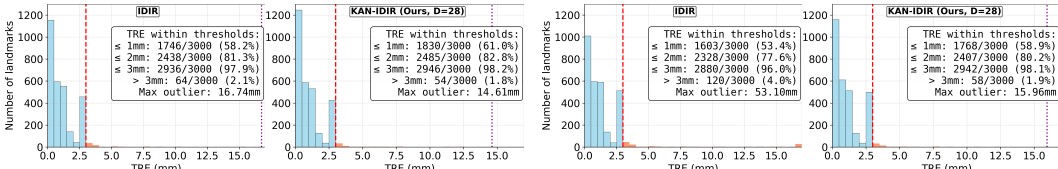

Figure 1: Target registration error (TRE) distributions for IDIR and proposed KAN-IDIR on DIR-Lab dataset. First two histograms show results for each method's best seed (lowest mean TRE), and last two histograms for the worst seed (highest mean TRE) out of 10 runs. The dashed red line marks the 3 mm threshold; the dotted purple line marks maximum outlier. KAN-IDIR produces fewer $> 3$ mm outliers under the best-performing seed (54, compared to 64 for IDIR) and also exhibits substantially lower sensitivity to the choice of seed. Specifically, outlier count for IDIR increases from 64 (best seed) to 120 (worst seed), a difference of 56, whereas KAN-IDIR increases only from 54 to 58, a difference of 4. This large contrast indicates that KAN-IDIR is considerably more robust to initialization.

sical iterative methods – an essential advantage for real-time clinical applications. However, such learning-based methods face three primary limitations: (1) their performance relies on large training datasets, which are often unavailable for specific organs or imaging modalities; (2) they do not consistently outperform classical methods for certain anatomical structures; and (3) they exhibit a degradation of registration quality when input data resolution varies. An alternative approach to dataset-trained registration methods is to learn the deformation field as a mapping from continuous voxel coordinates to the corresponding displacement vector. This mapping is parametrized by a neural network, referred to as an implicit neural representation (INR). INRs have demonstrated remarkable performance in tasks like novel view synthesis (NeRFs) (Mildenhall et al., 2020) and have been leveraged in several methods for DIR (Wolterink et al., 2022; Harten et al., 2023; 2024).

INR-based solutions address the main limitations of dataset-trained deep learning models: their reliance on large training datasets, sensitivity to domain shifts, and difficulties with handling substantially different resolutions. However, these methods require pairwise optimization, which makes computational efficiency and seed-dependent learning stability crucial for the practical use of these methods. Most existing INR architectures are based on multi-layer perceptrons (MLPs) with various activation functions (Sitzmann et al., 2020; Saragadam et al., 2023; Liu et al., 2024a). Recently, Kolmogorov-Arnold Networks (KANs), with their learnable activations, have emerged as a promising alternative to traditional MLPs in regression tasks (Liu et al., 2024c). Several recent studies have examined the INRs based on KANs across various tasks (Mehrabian et al., 2024; Li et al., 2025). To date, however, no prior work has investigated KAN-based INRs for deformable image registration, leaving their potential to advance registration techniques unexplored.

In this paper, we propose a novel approach to DIR using Implicit Neural Representations enhanced with Kolmogorov-Arnold Networks. Our key contributions are:

1. We introduce the first integration of Kolmogorov-Arnold Networks with randomized basis sampling for INR-based deformable image registration. Our approach enhances registration accuracy while maintaining low computational and memory costs compared to classical or neural network-based methods. Additionally, it demonstrates improved seed-dependent learning stability compared to other INRs (see Figure 1).

2. Our models, namely KAN-IDIR and RandKAN-IDIR, are the only INR-based solutions that achieve the best or best-in-class performance across all three evaluated datasets spanning different organs and imaging modalities. Specifically, they deliver superior accuracy and runtime/memory efficiency on DIR-Lab (lung CT), surpass other pairwise solutions on OASIS-1 (brain MRI), and achieve the leading perfomance and best deformation regularity on ACDC (cardiac MRI). Besides that, the proposed models result in fewer outliers compared to the other INR-based baseline method for the DIR-Lab dataset (see Figure 1).

3. Surprisingly, our RandKAN-IDIR model with randomized basis sampling achieved similar performance to the adaptive KAN variant that learns function indices, while eliminating their additional computational overhead during training.

## 2 RELATED WORKS

**Implicit Neural Representations.** Implicit neural representations model signals as continuous, differentiable functions parameterized by neural networks. Following (Sitzmann et al., 2020), an INR is defined as the solution to an equation:

$$\mathcal{C}(\mathbf{x}, \Phi, \nabla_{\mathbf{x}}\Phi, \nabla_{\mathbf{x}}^2\Phi, \ldots) = 0, \quad \Phi : \mathbf{x} \mapsto \Phi(\mathbf{x}). \tag{1}$$

Here, a network is trained to parameterize the function $\Phi$ that maps coordinates $\mathbf{x} \in \mathbb{R}^n$ to target values while satisfying the implicit constraint $\mathcal{C}$ in Eq. 1.

The choice of INR architecture is critical, as it determines the class of functions that the network can effectively represent. While ReLU-based MLPs mainly model low-frequency signals (Mildenhall et al., 2020), many works explore alternative activation functions — such as the sine functions in SIREN (Sitzmann et al., 2020), Gaussian activations (Ramasinghe & Lucey, 2022) , wavelet-based activations in WIRE (Saragadam et al., 2023), or variable-periodic functions in FINER (Liu et al., 2024a) — to better capture high-frequency structure. These designs, however, introduce their own challenges. For example, the expressiveness of SIREN depends heavily on frequency-related hyperparameters and is sensitive to initialization, requiring careful tuning to avoid unstable or inconsistent behavior (Vonderfecht & Liu, 2024). Moreover, instead of relying on hand-crafted periodic or frequency-aware activations, recent work has explored learnable activation functions, whose parameters adapt during training (Goyal et al., 2019). Spline-based versions, such as those used in KAN (Liu et al., 2024c), offer a flexible way to model nonlinearities and frequency content, helping mitigate some limitations of fixed activation choices. Nevertheless, SIREN remains a strong and widely used baseline in many domain-specific applications, including medical image registration.

**Medical Image Registration.** Classical iterative image registration methods are built upon well-established and rigorously developed mathematical frameworks. (Rueckert et al., 1999) introduced a non-rigid registration approach using B-spline-based free-form deformations (FFD) combined with an affine global transformation and normalized mutual information to model complex local tissue motion. LDDMM (Beg et al., 2005) models large, smooth deformations by interpreting them as geodesic flows, grounded in concepts from fluid mechanics and Lagrangian dynamics. (Thirion, 1998) proposed a model in which one image diffuses through the boundaries of another, analogous to Maxwell's demons (Maxwell, 1871). The pTV method (Vishnevskiy et al., 2016) parameterizes the displacement field with B-splines and incorporates isotropic total variation regularization to better capture breathing motion. Flash (Zhang & Fletcher, 2019) accelerates diffeomorphic registration by representing the initial velocity field in a low-dimensional Fourier space, reducing computational cost with minimal loss of accuracy. While these classical approaches achieve high accuracy through per-case optimization, they are often computationally intensive – particularly for high-resolution images – and require extensive hyperparameter tuning to reach optimal performance. These factors limit their scalability and practicality in clinical settings.

Recent advances in deep learning (DL) for medical image registration have been heavily CNN-based methods. The VoxelMorph framework (Balakrishnan et al., 2019a) was a major breakthrough, enabling fast deformable registration using CNNs to learn deformation fields in unsupervised manner. It was further extended with probabilistic modeling with stationary velocity fields (SVFs) (Dalca et al., 2019), diffeomorphic networks in LapIRN approach (Mok & Chung, 2020b), and methods ensuring symmetry and topology preservation (Mok & Chung, 2020a; Greer et al., 2021; Tian et al., 2023). The rapid development of vision transformers introduced several strong approaches for DIR that utilized the ability of transformers to extract more complex spatial patterns from data that resulted in better registration quality (Chen et al., 2022). Beyond CNNs and transformers, (Meng et al., 2024) developed a pure-MLP method, achieving top results on some datasets.

While CNN-based medical image registration has advanced significantly, key limitations persist. First, even when trained on large datasets, these image-to-image models remain vulnerable to domain shifts (Jena et al., 2024) - a problem often addressed through even larger datasets (Tian et al., 2024). Second, while segmentation labels enhance performance in brain MRI (Jena et al., 2024), these methods underperform on unlabeled data (e.g., DIR-Lab lung CT (Castillo et al., 2009)), trailing classical (Vishnevskiy et al., 2016) and INR-based (Harten et al., 2023) approaches. Finally, the CNNs and ViT by design typically cannot infer on several times higher or lower resolution without

retraining, a limitation partially resolved by resolution-robust neural operators (Drozdov & Sorokin, 2024) at the cost of slightly reduced accuracy.

To the best of our knowledge, there is no established universal SOTA method for medical image registration as a whole. While a few recent foundation-style approaches aim to build general-purpose DIR models (Tian et al., 2024), state-of-the-art performance is still defined on a per-dataset basis — for example, the leading results on DIRLab (Vishnevskiy et al., 2016), OASIS (Chen et al., 2022; Jena et al., 2024), and ACDC (Meng et al., 2024).

**INRs for Image Registration.** In INR-based registration, the deformation field $\Phi$ is modeled as a continuous neural function over the spatial domain $\Omega$, fundamentally differing from discrete image-to-image approaches. While INRs require optimizing a new network for each image pair, making it more computationally demanding than a single forward pass through a pre-trained CNN or transformer, it offers significant advantages. First, by eliminating the need for pretraining on large dataset, the INR-based methods require significantly less computational resources. Second, pair-specific registration algorithms also inherently avoid domain shift issues. Finally, the continuous nature of INRs makes them resolution-independent, allowing seamless application to data at arbitrary resolutions without architectural modifications.

INR-based medical image registration was pioneered by (Wolterink et al., 2022) using SIRENs for lung CT, achieving performance competitive with both classical and learning-based methods. Subsequent work extended this to brain MRI with alternative activations and cycle consistency (Byra et al., 2023), and later to robust bidirectional frameworks with uncertainty quantification (Harten et al., 2023). Another pairwise optimization approach (but not explicitly INR) reformulated registration via neural ODEs in NODEO (Li et al., 2021). Most recently, (Sideri-Lampretsa et al., 2024) hybridized B-spline FFDs with INRs.

As demonstrated in our experiments below, current methods struggle to achieve an optimal balance between essential DIR requirements, including registration accuracy and deformation smoothness, as well as INR-specific constraints arising from the need of per-case network training, such as computational efficiency, memory demands, and seed-dependent stability.

**KANs.** Modern deep learning relies heavily on MLPs, known for their universal approximation capability (Cybenko, 1989; Hornik et al., 1989). The Kolmogorov-Arnold approximation theorem for multivariate continuous functions (Kolmogorov, 1956; Arnold, 1957) offers an alternative: any continuous $f : [0,1]^n \to \mathbb{R}$ can be expressed as

$$f(x_1, x_2, \ldots, x_n) = \sum_{q=1}^{2n+1} \Phi_q \left( \sum_{p=1}^{n} \phi_{q,p}(x_p) \right), \qquad (2)$$

where $\phi_{q,p} : [0,1] \to \mathbb{R}$ and $\Phi_q : \mathbb{R} \to \mathbb{R}$ are continuous univariate functions.

The authors of (Liu et al., 2024c) introduced shallow and deep Kolmogorov-Arnold Networks (KANs) based on (2), where each $\phi_{q,p}$ is implemented as spline-based univariate functions. While theoretically universal, shallow KANs with limited basis functions face convergence issues. Deep KAN architectures with multiple layers address this issue.

Recent work has explored KANs as INRs across domains: (Mehrabian et al., 2024) first adapted KAN layers for standard INR tasks (image fitting, occupancy volumes), while (Li et al., 2025) developed Fourier-KAN for neural sound fields. Several studies (Liu et al., 2024b; Faroughi & Mostajeran, 2025) have demonstrated that KANs outperform MLPs in accurately approximating functions of a physical or mathematical nature, like special functions or solutions to partial differential equations. Image registration can be formulated as finding the solution to an Euler-Lagrange system of a specific target functional (Drozdov & Sorokin, 2024), providing strong justification for using KANs in this context.

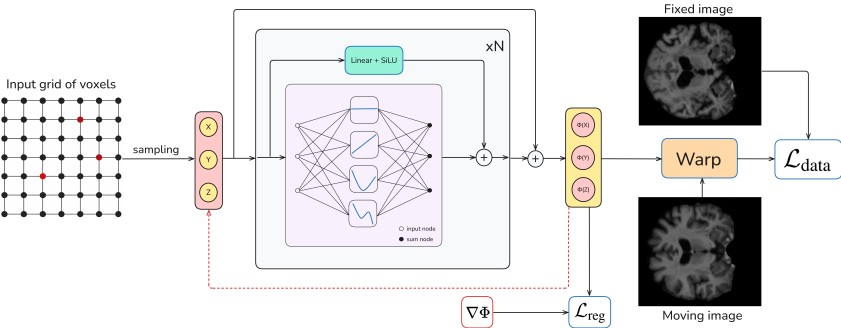

Figure 2: The overall pipeline of the proposed method for deformable image registration. Voxels are sampled from the input volume to serve as input to the KAN-IDIR or RandKAN-IDIR models. The network outputs a deformation field, which is used to warp the moving image, enabling calculation of the data similarity loss. For regularization, the gradient of the deformation field is computed using automatic differentiation.

## 3 METHODOLOGY

### 3.1 IMAGE REGISTRATION WITH INRS

DIR problem can be considered as finding the deformation field $\Phi(x) = x + U(x)$ that minimizes the following objective:

$$\hat{\Phi} = \arg\min_{\Phi} \mathcal{L}_{\text{data}}(M \circ \Phi, F) + \mathcal{L}_{\text{reg}}(\Phi), \tag{3}$$

where $\mathcal{L}_{\text{data}}$ is a similarity measure between the fixed image $F$ and the warped moving image $M \circ \Phi$, and $\mathcal{L}_{\text{reg}}$ is a regularization term imposing constraints on the deformation field $\Phi$. We assume that the fixed and moving images are defined on the same domain, i.e., $M, F : \Omega \subset [-1, 1]^n \to \mathbb{R}$.

### 3.2 INR ARCHITECTURE

We adopt the Chebyshev KAN (Sidharth et al., 2024) as architecture of INR. This network utilizes Chebyshev polynomials with a fixed grid as basis functions, in contrast of B-splines with an adaptive grid used in the original KAN (Liu et al., 2024c). Chebyshev polynomial of the $n^{\text{th}}$ order is defined analytically as

$$T_n(x) = \cos(n \cdot \arccos(x)), \ x \in [-1, 1]. \tag{4}$$

Using this definition, we can perform efficient vectorized computations of $T_n$ values, thus increasing the network's speed in comparison to the original B-spline representation. Forward pass through the Chebyshev KAN's layer can be mathematically defined as follows:

$$[\mathbf{y}(\mathbf{x})]_{b,o} = \sum_{i=1}^{N_{in}} \sum_{d \in \mathcal{D}} T_d(\mathbf{x}_{b,i}) \cdot \mathbf{C}_{i,o,d}, \tag{5}$$

where $b$ indexes the samples in the data batch, $i$ indexes the input dimension, $o$ indexes the output dimension, and $d$ indexes the polynomial degree. Here, $\mathbf{x}$ represents the input tensor, $\mathbf{y}$ the output tensor, $\mathbf{C}$ denotes the tensor of learnable coefficients, and $\mathcal{D} \subset \mathbb{N}_0$ is a finite set of all basis polynomial degrees, which may be shared across layers or vary between them.

We use normalization with the hyperbolic tangent function (tanh) to map input tensor values in $[-1, 1]$ interval before passing into Chebyshev polynomials. To stabilize the training process, we follow the approach of the original KAN model and incorporate a learnable skip-connection in each layer. More precisely, the forward pass through $m^{\text{th}}$ layer of our network is determined by the following expression:

$$L_m(\mathbf{x}) = W_m b(\mathbf{x}) + \mathbf{y}_m(\widetilde{\mathbf{x}}), \tag{6}$$

where $\mathbf{x}$ is the input tensor, $\widetilde{\mathbf{x}}$ denotes the inputs after tanh normalization, $\mathbf{y}_m(\cdot)$ represents the nonlinear part as defined in Eq. 5 and $W_m$ is the learnable matrix of linear transformation. We have chosen $b(\mathbf{x}) = \text{silu}(\mathbf{x}) = x/(1 + e^{-x})$ as an activation function for linear skip-connection.

Using the definitions above, the architecture of our model can be written as the composition of $N$ layers from Eq. 6:

$$\Phi(\mathbf{x}) = (L_{N-1} \circ L_{N-2} \circ \ldots \ L_0)(\mathbf{x}), \ \mathbf{x} \in \Omega \,. \tag{7}$$

### 3.3 KANs with Sparse Set of Basis Functions

Sparse approximations were among the most prominent concepts in signal processing and pattern recognition during the pre-deep learning era (Zhang et al., 2015). The core idea of these algorithms is that a target signal can be approximately represented by a sparse linear combination of prototype signals from an overcomplete dictionary (Mallat & Zhang, 1993; Aharon et al., 2006). Inspired by the successful application of sparse representations in the past, we explored whether adopting a similar approach for KANs could enhance the quality and efficiency when applying to the DIR task. In the case of the KAN model, the set of basis functions serves as an overcomplete dictionary, and our goal is to utilize only a few of them during each forward pass of data through the network. Thus, two strategies can be considered: either selecting a *fixed* sparse set of basis functions for each layer before training or learning the optimal selection *dynamically* during network optimization. Based on these two strategies together with classical sequential selection of basis functions, we evaluate three distinct models in our subsequent work:

**KAN-IDIR**: This model employs the classical basis selection scheme with a fixed maximum degree $D$. For each layer, the basis functions are polynomials with degrees from $0$ to $D$, i.e., $\mathcal{D} = \{0, 1, \ldots, D\}$. The set $\mathcal{D}$ of polynomial degree indices is shared across all layers.

**RandKAN-IDIR**: This model implements basis selection with randomized sampling. Let $\mathcal{D}_m$ denote the set of degrees of the basis polynomials for the $m^{\text{th}}$ layer. It is constructed by including the constant term (degree 0) and combining it with a random selection of higher degrees: $\mathcal{D}_m = \{0\} \cup \mathcal{S}_m$, where $\mathcal{S}_m$ is a simple random sample without replacement of size $k$ from the set $\{1, 2, \ldots, K\}$, independently for each layer $m$. Despite the reduced basis set, experiments (Sec. 4) show that the RandKAN-IDIR model maintains competitive registration accuracy while improving efficiency, given an appropriate choice of $k$ and $K$.

**A-KAN-IDIR**: Adaptively learns basis indices during training (see Section A.1 in Appendix for details). Experiments show no quality improvement over RandKAN-IDIR's random selection, while being computationally slower, so we excluded this model from evaluations.

### 3.4 Training Objective

Our loss function combines similarity and regularization terms (Eq. 3). For the similarity term, we use negative normalized cross-correlation (NCC):

$$\mathcal{L}_{\text{data}}(I, J) = -NCC(I, J) = -\frac{\sum_{\mathbf{p} \in \Omega_p} (I(\mathbf{p}) - \bar{I})(J(\mathbf{p}) - \bar{J})}{\sqrt{\sum_{\mathbf{p} \in \Omega_p} (I(\mathbf{p}) - \bar{I})^2 \sum_{\mathbf{p} \in \Omega_p} (J(\mathbf{p}) - \bar{J})^2}},$$

where $\Omega_p \subset \Omega$ is the current batch of voxel coordinates, and $\bar{I}, \bar{J} \in \mathbb{R}$ are the mean intensities of corresponding images over $\Omega_p$.

The regularization term $\mathcal{L}_{\text{reg}}$ combines two components: a smoothness term $\mathcal{L}_{\text{smooth}}$ and a folding prevention term $\mathcal{L}_{\text{Jdet}}$, weighted by parameters $\lambda$ and $\gamma$ respectively. To enforce physically plausible deformations, we employ total variation (TV) regularization (Rudin et al., 1992), which penalizes the L1 norm of the displacement field gradient:

$$\mathcal{L}_{\text{smooth}} = \frac{1}{|\Omega_p|} \sum_{\mathbf{p} \in \Omega_p} ||\nabla U(\mathbf{p})||_1. \tag{8}$$

While total variation (TV) regularization effectively promotes smooth deformation fields, it may still permit folded voxels, i.e., voxels with locally non-diffeomorphic transformations characterized by negative Jacobian determinants ($|J_\Phi| < 0$). To address this, we incorporate a Jacobian determinant regularization term (Mok & Chung, 2020a) that actively penalizes folding regions:

$$\mathcal{L}_{\text{Jdet}} = \frac{1}{|\Omega_p|} \sum_{\mathbf{p} \in \Omega_p} \sigma(-|J_\Phi(\mathbf{p})| + \epsilon), \tag{9}$$

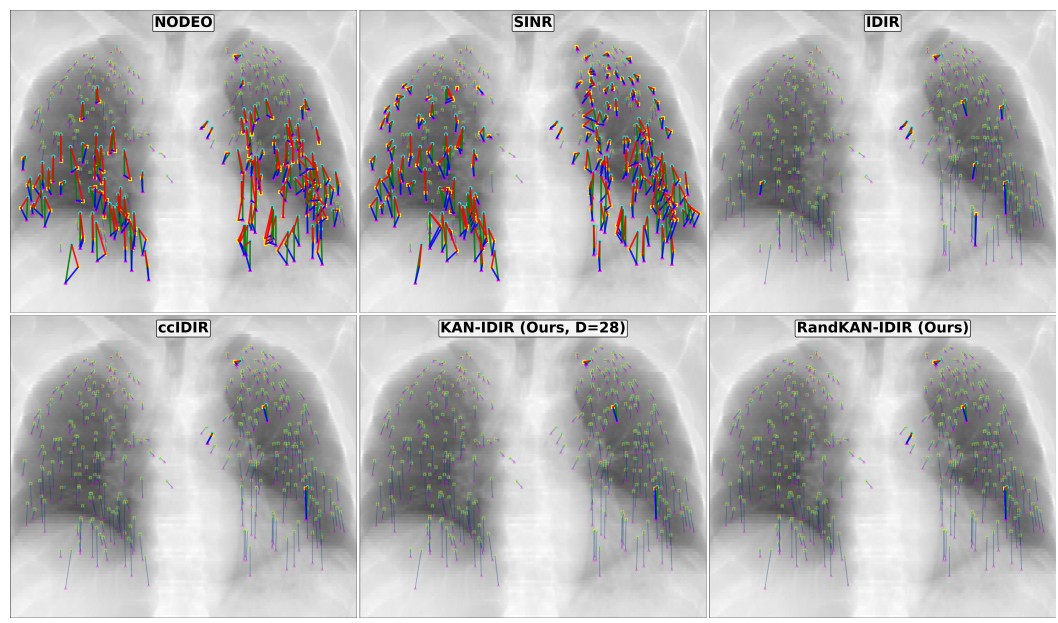

Figure 3: Landmark alignment in DIR-Lab Dataset (Case 7) for fixed, moving, and moved images. Displacements are shown as: blue (method result, moved-to-fixed), green (ground truth, moving-to-fixed), and red (error, moved-to-fixed). Landmarks with errors exceeding 3 mm are highlighted in bold, while others are displayed as transparent. Results reflect each method's optimal seed (minimum mean TRE across 10 runs). Proposed KAN-IDIR and RandKAN-IDIR achieve the fewest outliers. The best seed-dependent variability of the proposed methods is demonstrated in Figure 7 in Appendix using worst-case (maximum mean TRE) results.

where $\sigma(x) = \max(0, x)$ is the ReLU function and $\epsilon > 0$ is a small constant added for overcorrection (Li et al., 2021). The complete training objective combines these regularization terms with the data similarity term:

$$\mathcal{L} = \mathcal{L}_{data} + \lambda \mathcal{L}_{\text{smooth}} + \gamma \mathcal{L}_{\text{Jdet}}. \tag{10}$$

The overall pipeline of the proposed method for deformable image registration is shown in Figure 2.

## 4 EXPERIMENTS

### 4.1 DATASETS AND EVALUATION METRICS

We evaluated our approach using three established DIR datasets spanning three organs and two imaging modalities.

**DIR-Lab** (Castillo et al., 2009): Ten lung 4D CT scans (inspiration/expiration phases) with 300 human-identified landmarks per pair corresponding to stable anatomical features, capturing complex cardiopulmonary motion posed by the interplay of cardiac and respiratory movements, which significantly exceed the scale of small lung structures. Image resolutions: $256^2$-$512^2$ in-plane, 94-136 Z-axis slices.

**OASIS-1** (Marcus et al., 2007): 414 affinely-aligned brain MRIs with segmentations of 35 major anatomical regions, preprocessed to $160\times192\times224$ size (Hoopes et al., 2022). We followed Learn2Reg (Hering et al., 2022) evaluation protocols, using 49 test pairs from last 50 subjects for evaluating the compared approaches.

**ACDC** (Bernard et al., 2018): 150 cardiac MRI pairs (ED/ES phases) with ventricular segmentations. We evaluated on 50 pairs registered bidirectionally (100 pairs total), preprocessed to $128\times128\times32$ size in accordance with CorrMLP (Meng et al., 2024).

Table 1: Evaluation of baselines and proposed methods (in **bold**) on the DIR-Lab dataset. The best result is in **bold**, and the second-best is in *italics*. Our models achieve near state-of-the-art performance in lung CT registration with lowest execution times.

| Methods | pTV | NODEO | SINR | IDIR | ccIDIR | **KAN-IDIR (28)** | **RandKAN-IDIR** |
|---|---|---|---|---|---|---|---|
| Average TRE, mm ↓ | **0.95 (1.15)** | 3.93 (6.07) | 4.63 (5.26) | 1.07 (1.10) | 1.04 (1.12) | *0.98 (1.10)* | 0.99 (1.10) |
| NJD, % ↓ | 0.6 (–) | **0.0003 (0.0003)** | 0.002 (0.002) | 0.002 (0.015) | 0.002 (0.008) | *0.0006 (0.001)* | 0.006 (0.016) |
| Runtime, sec ↓ | 442 | 583.5 | 91.6 | 260.8 | 92.6 | *63.3* | **43.1** |
| VRAM, Gb ↓ | – | 18.7 | 21.7 | 4.1 | **1.0** | 2.2 | *1.4* |

Table 2: Cumulative distribution of TRE for DIR-Lab data shown as percentages within thresholds ($\leq$ 1-3mm) and for outliers ($>$ 3mm) across pairwise DL methods. Proposed method names and best values in **bold**.

| Threshold | NODEO | SINR | IDIR | ccIDIR | **KAN-IDIR (28)** | **RandKAN-IDIR** |
|---|---|---|---|---|---|---|
| $\leq$1mm ↑ | 39.05% | 18.14% | 56.51% | 57.07% | *60.13%* | **60.28%** |
| $\leq$2mm ↑ | 57.23% | 34.56% | 79.78% | 79.97% | **81.69%** | *81.43%* |
| $\leq$3mm ↑ | 73.09% | 55.59% | 97.52% | 97.95% | **98.14%** | *98.09%* |
| $>$3mm ↓ | 26.91% | 44.41% | 2.48% | 2.05% | **1.86%** | *1.91%* |

For all datasets, we evaluated four **quantitative metrics**: registration accuracy, measured by Target Registration Error (TRE) for DIR-Lab and by Dice score (DSC) and 95[th] percentile Hausdorff distance (HD95) for OASIS-1 and ACDC; deformation regularity, quantified as the percentage of folded voxels (NJD); GPU memory consumption; and runtime. Experiments ran on an Intel Xeon Gold 6226R CPU / NVIDIA A6000 server under exclusive load to ensure runtime precision measurement. For INR-based methods, metrics were averaged over ten random seeds (mean±std), following (Harten et al., 2023; Wolterink et al., 2022).

## 4.2 IMPLEMENTATION DETAILS

We implemented our method using the PyTorch library (Paszke et al., 2019). All KAN-based models consist of two hidden layers, each with 70 neurons. This configuration was chosen to approximately match the number of parameters in our models to that of classical INR-based registration approaches (Wolterink et al., 2022; Harten et al., 2023). For the parameters $\lambda$, $\gamma$ and $\epsilon$ in Eq. 9-10, we adopt the values $\lambda = 0.4$, $\gamma = 15$ and $\epsilon = 0.1$ unless otherwise specified. The Adam optimizer (Kingma & Ba, 2014) was employed for model training with an initial learning rate of $10^{-4}$. The learning rate remained constant for the first 50% of the iterations, after which it decreased following a cosine annealing schedule. Each model was trained for a total of 1500 iterations, with 10,000 points randomly sampled from the domain of interest on each iteration. The domain of interest is defined by a segmentation mask that separates the organ from the background. For the DIR-Lab dataset, we followed the approach from (Wolterink et al., 2022) and used the pretrained segmentation model to obtain the lung mask (Hofmanninger et al., 2020). The OASIS-1 dataset provides reference masks for anatomical segments, enabling us to use the union of all non-background segmentations as the resulting mask. For the ACDC dataset, we used all image voxels for training. For comprehensive baseline implementation details we refer readers to the Appendix (Section B.1).

## 4.3 RESULTS

Our evaluation compares against: neural ODE-based NODEO (Li et al., 2021); INR-based methods (IDIR (Wolterink et al., 2022), ccIDIR (Harten et al., 2023), SINR (Sideri-Lampretsa et al., 2024)); dataset-specific SOTA (pTV (Vishnevskiy et al., 2016) for DIR-Lab, CorrMLP (Meng et al., 2024) for ACDC); and established DIR baselines (VoxelMorph (Balakrishnan et al., 2019a), TransMorph (Chen et al., 2022) for OASIS-1).

We denote KAN-IDIR ($D$) as the model configuration with a maximum degree of $D$. For the RandKAN-IDIR model, we use a configuration with $k = 12$ and $K = 84$ in all experiments unless otherwise specified.

Table 3: Results of experiments on the OASIS-1 dataset. For learning-based methods, runtime and memory consumption are reported for both inference and training in the format inference/training. Our KAN-IDIR and RandKAN-IDIR hit the highest registration quality among INR-based methods, with nearly the fastest execution times and low VRAM consumption. Proposed method names and best values in **bold**, second-best in *italics*.

| Methods | VM | TM | NODEO | SINR | IDIR | ccIDIR | KAN-IDIR (28) | RandKAN-IDIR |
|---|---|---|---|---|---|---|---|---|
| DSC ↑ | *0.795 (0.02)* | **0.811 (0.02)** | 0.790 (0.17) | 0.729 (0.04) | 0.762 (0.18) | 0.779 (0.16) | **0.793 (0.16)** | *0.792 (0.17)* |
| HD95 ↓ | *1.94 (0.29)* | **1.81 (0.30)** | 1.98 (0.38) | 2.65 (0.47) | 2.22 (0.45) | 2.19 (0.44) | **1.96 (0.32)** | *1.97 (0.32)* |
| NJD, % ↓ | *0.564 (0.11)* | 0.526 (0.14) | **0.010 (0.01)** | *0.020 (0.01)* | 0.042 (0.03) | 0.315 (0.12) | 0.048 (0.03) | 0.049 (0.04) |
| Runtime ↓ | 0.1 s / 2.1 days | 0.2 s / 3.7 days | 68.1 s | **20.6 s** | 260.8 s | 92.6 s | 63.3 s | *43.1 s* |
| VRAM, Gb ↓ | 8.7 / 14.1 | 19.8 / 22.0 | 2.7 | 17.5 | 4.4 | **1.0** | 2.2 | *1.4* |

Table 4: Results of experiments on the ACDC dataset. Values marked with an asterisk (*) are taken directly from (Meng et al., 2024). KAN-based models achieve competitive performance with superior deformation regularity. Proposed method names and best values in **bold**, second-best in *italics*.

| Methods | CorrMLP | NODEO | SINR | IDIR | ccIDIR | KAN-IDIR (8) | KAN-IDIR (28) | RandKAN-IDIR |
|---|---|---|---|---|---|---|---|---|
| DSC ↑ | 0.810 | 0.795 (0.095) | *0.811 (0.07)* | **0.814 (0.09)** | 0.792 (0.093) | **0.814 (0.09)** | 0.809 (0.09) | 0.810 (0.09) |
| HD95 ↓ | – | 7.87 (3.17) | **6.75 (3.11)** | *7.38 (3.38)* | 8.23 (3.15) | 7.39 (3.17) | 7.87 (3.25) | 7.88 (3.28) |
| NJD, % ↓ | 0.389 | 0.011 (0.045) | 0.002 (0.002) | 0.111 (0.166) | 0.051 (0.066) | **2e-5 (2e-4)** | 5e-5 (4e-4) | *4e-5 (2e-4)* |
| Runtime, s ↓ | 0.07* | *19.6* | **13.9** | 260.8 | 92.6 | 40.6 | 63.3 | 43.1 |
| VRAM, Gb ↓ | - | **0.7** | 2.6 | 4.4 | *1.0* | 1.1 | 2.2 | 1.4 |

**Results on DIR-Lab Dataset.** Table 1 presents the experimental results obtained on the DIR-Lab dataset. KAN-IDIR and RandKAN-IDIR achieve the second and third best TRE, respectively, outperformed only by the classical pTV method, which remains SOTA for this dataset against all classical and DL-based approaches. However, our methods are significantly faster: RandKAN-IDIR registers pairs in under one minute versus pTV's several minutes. Additionally, the performance difference between RandKAN-IDIR and KAN-IDIR is minimal, with only 0.01 mm average TRE gap, while RandKAN-IDIR achieves approximately 30% faster execution (43.1 s versus 63.3 s). Notably, while NODEO and SINR perform competitively on other modalities, they show poor lung CT performance with TRE $\approx 4$ times higher than our methods.

Figure 1 depicts the distribution of registration errors across all landmark pairs in the DIR-Lab dataset, while Table 2 presents the percentage of errors within specified thresholds. It demonstrates that our methods not only achieve superior average registration quality but also exhibit improved quality in terms of error distribution and seed-dependent variability. Figure 3 visualizes the overlay of the human-defined landmarks used to calculate TRE showing the qualitative behavior of the evaluated INR-based methods.

**Results on OASIS-1 Dataset.** For the OASIS-1 dataset, we also include the learning-based methods such as VoxelMorph and TransMorph in the comparison. Table 3 shows the results of experiments on the OASIS-1 dataset. While learning-based methods outperform case-specific approaches in registration quality, they demand substantial computational resources for training ($10^6 \times$ greater training time and $10 \times$ higher VRAM consumption on both training and inference). Compared to other INR-based methods, KAN-IDIR and RandKAN-IDIR achieve the best and second-best results in terms of DSC and HD95, while maintaining a moderate level of folded voxels in the deformation field. Furthermore, among INR-based methods, our models achieve nearly the fastest runtimes and memory consumption of no more than 2.2 GB, enabling their execution on consumer-grade GPUs. The visual results for the OASIS-1 dataset are provided in the Appendix (Figure 5).

**Results on ACDC Dataset.** Table 4 presents the experimental results for the ACDC dataset. Most methods exhibit similar performance levels, which we attribute to the relatively small number of voxels available for optimization, promoting a more stable training process. The proposed KAN-IDIR model with $D = 8$ shares the highest DSC value with IDIR and achieves the absolute best NJD result – only 0.00002% of foldings. Our DSC value overcomes the current state-of-the-art DSC reported in (Meng et al., 2024). Similarly, RandKAN-IDIR provides competitive registration

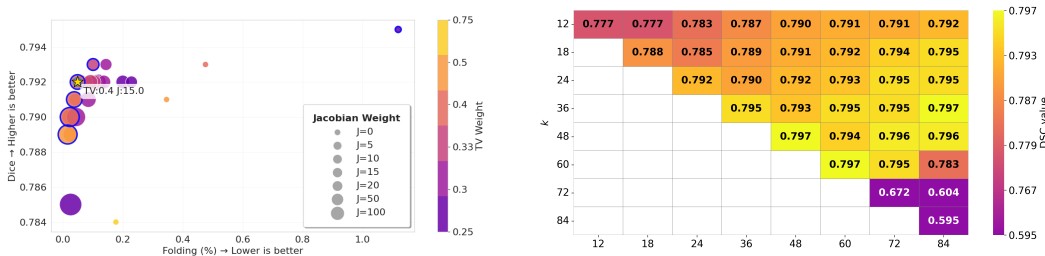

Figure 4: **Left:** Ablation study of loss weights for RandKAN-IDIR model on the OASIS-1 dataset. Configurations on the Pareto frontier are highlighted in blue. **Right:** Ablation study of $k$ and $K$ parameters for RandKAN-IDIR model on the OASIS-1 dataset.

quality while exhibiting excellent deformation field regularity. The visual comparison for the ACDC dataset is presented in the Appendix (Figure 6).

### 4.4 Ablation Studies and Parameter Tuning

**Loss Function Weights.** In this study, we emphasize the importance of Jacobian determinant regularization in our method. Figure 4 (left) illustrates the results for the RandKAN-IDIR model on the OASIS-1 dataset across 26 different loss function weight configurations. Combinations with $\gamma = 0$ (smallest circles) exhibit poorer deformation regularity compared to others, even when increasing the $\lambda$ weight of the TV regularizer. Our selected configuration, with $\lambda = 0.4$ and $\gamma = 15$, lies on the Pareto frontier, achieving a balance between Dice score and the amount of foldings.

**Choice of $k$ and $K$ parameters.** Figure 4 (right) shows the results for the RandKAN-IDIR model on the OASIS-1 dataset with different values of $k$ and $K$. Overall, performance improves monotonically with increasing $k$ up to 60, beyond which the model encounters convergence issues. For a fixed $k$, models with randomized sampling (results for $K > k$) and sufficiently large value of $K$ achieve better registration quality than models with sequential basis selection (results on the main diagonal with $K = k$). The improvement is especially notable for smaller values of $k$. For example, the model with $k = 12$ and $K = 84$ outperforms the model with $k = K = 12$ (equivalent to KAN-IDIR with $D = 12$) by 1.5% DSC and matches the performance of the model with $k = K = 24$. Moreover, our models can be scaled to reach slightly higher performance than reported in Table 3 – up to 0.797 DSC – at the cost of slower runtime and increased VRAM consumption.

## 5 Conclusion

This paper presents KAN-IDIR and RandKAN-IDIR, the first models to incorporate KANs into INR-based deformable image registration. By integrating randomized basis sampling into KAN-IDIR, RandKAN-IDIR reduces computational complexity while preserving near-equivalent registration quality. Both models deliver best or best-in-class performance across three medical imaging datasets of different organs and modalities, achieving superior deformation regularity and minimal computational overhead compared to other INR-based methods. Additionally, KAN-IDIR and RandKAN-IDIR demonstrate improved seed-dependent stability compared to their counterparts, ensuring consistent performance across diverse scenarios.

Looking ahead, our framework enables new research directions in KAN architecture optimization for broader medical image analysis tasks. The method's computational efficiency, seed-dependent stability, and instance-specific adaptability make it particularly suitable for clinical adoption, especially when working with limited datasets.

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

# A  APPENDIX: ADAPTIVE LEARNING OF BASIS FUNCTIONS

## A.1  DESCRIPTION OF A-KAN-IDIR VS RANDKAN-IDIR

**A-KAN-IDIR architecture.** Here we describe our A-KAN-IDIR model with adaptive learning of basis function indices.

We assign a learnable weight to each function in the basis function set, with each weight reflecting the importance of its corresponding basis function. The forward pass through the A-KAN-IDIR layer is formulated as follows:

$$[\mathbf{y}(\mathbf{x})]_{b,o} = \sum_{i=1}^{N_{in}} \sum_{d \in \mathcal{D}_w} \text{sigmoid}(w_d) \cdot T_d(\mathbf{x}_{b,i}) \cdot \mathbf{C}_{i,o,d}, \tag{11}$$

where $w_d$ represents the learnable weight, constrained to $(0, 1)$ with the sigmoid function, and $\mathcal{D}_w$ denotes the set of indices for basis functions selected at each iteration. To construct $\mathcal{D}_w$, we employ

Table 5: Comparison of A-KAN-IDIR and RandKAN-IDIR on DIR-Lab, OASIS-1, and ACDC datasets. The best result per metric is highlighted in **bold**.

| Method | DIR-Lab | | OASIS-1 | | | ACDC | | |
|---|---|---|---|---|---|---|---|---|
| | Avg. TRE ↓ | NJD ↓ (%) | DSC ↑ | HD95 ↓ | NJD ↓ (%) | DSC ↑ | HD95 ↓ | NJD ↓ (%) |
| A-KAN-IDIR | 1.02 | **0.006** | 0.786 | 2.01 | 0.135 | **0.811** | **7.76** | 8e-5 |
| RandKAN-IDIR | **0.99** | **0.006** | **0.792** | **1.97** | **0.049** | 0.810 | 7.88 | **4e-5** |

the Noisy Top-$k$ algorithm, which selects the indices corresponding to the top-$k$ values of the noisy weight vector:

$$\mathcal{D}_w = \text{TopkIndices}(\{w_i + n_i \mid w_i \in \mathbf{w}\}), \tag{12}$$

where $\mathbf{w}$ denotes the vector of all weights, and $n_i$ represents Gaussian noise added to each weight $w_i$ to prevent convergence to local minima early in the training process. We model $n_i$ as Gaussian noise with a standard deviation that decreases from 0.3 at the start of training to zero by 75% of the iterations. During the final 25% of iterations, we freeze the learnable weights and train only the Chebyshev polynomial coefficients.

To achieve a richer parameterization of learnable basis function weights, we adopt the Deep Image Prior (Ulyanov et al., 2018) approach, where the target image is learned from a fixed noise vector. We learn the weight vector $\mathbf{w}$ for each layer from a unique fixed vector $\mathbf{z}$, sampled from a standard Gaussian distribution. The mapping from $\mathbf{z}$ to $\mathbf{w}$ is learned using a two-layer MLP. The values of $\mathbf{z}$, $\mathbf{w}$, and the MLP weights are layer-specific and not shared across layers.

The overall forward pass pipeline for A-KAN-IDIR layer is as follows:

1. Pass $\mathbf{z}$ to MLP and obtain the $\mathbf{w}$ values: $\mathbf{w} = \text{MLP}(\mathbf{z})$;

2. Sample noise vector $\mathbf{n}$;

3. Select set of indices $\mathcal{D}_w$ according to Eq 12;

4. Constrain values of $\mathbf{w}$ with indices from $\mathcal{D}_w$ with sigmoid function;

5. Perform forward pass according to Eq. 11.

Table 6: Evaluation of TRE ($\downarrow$) in mm for baselines and proposed methods on DIR-Lab dataset. Best result is highlighted in **bold**, second-best – in *italic*.

| Case | pTV | NODEO | SINR | IDIR | ccIDIR | KAN-IDIR (D=28, Ours) | RandKAN-IDIR (k=12, K=84, Ours) |
|------|-----|-------|------|------|--------|-----------------------|----------------------------------|
| 4DCT 01 | 0.76 (0.90) | 0.88 (0.94) | 1.59 (1.33) | 0.76 (0.94) | 0.83 (0.94) | 0.77 (0.91) | 0.78 (0.92) |
| 4DCT 02 | 0.77 (0.89) | 1.08 (1.62) | 1.81 (1.90) | 0.76 (0.94) | 0.78 (0.93) | 0.77 (0.93) | 0.76 (0.92) |
| 4DCT 03 | 0.90 (1.05) | 1.38 (2.13) | 3.36 (3.10) | 0.94 (1.02) | 1.02 (1.10) | 0.89 (1.02) | 0.91 (1.03) |
| 4DCT 04 | 1.24 (1.29) | 2.93 (3.64) | 3.82 (3.55) | 1.32 (1.27) | 1.37 (1.36) | 1.30 (1.28) | 1.29 (1.26) |
| 4DCT 05 | 1.12 (1.44) | 3.30 (4.53) | 4.76 (4.12) | 1.23 (1.47) | 1.25 (1.51) | 1.13 (1.41) | 1.13 (1.41) |
| ' 4DCT 06 | 0.85 (0.89) | 4.52 (5.55) | 4.58 (3.30) | 1.09 (1.03) | 1.06 (1.09) | 0.94 (1.01) | 0.94 (1.01) |
| 4DCT 07 | 0.80 (1.28) | 7.00 (7.93) | 6.74 (6.41) | 1.12 (1.00) | 0.97 (0.98) | 0.93 (0.96) | 0.94 (0.97) |
| 4DCT 08 | 1.34 (1.93) | 10.47 (10.34) | 12.10 (8.71) | 1.21 (1.29) | 1.13 (1.40) | 1.11 (1.26) | 1.13 (1.28) |
| 4DCT 09 | 0.92 (0.94) | 4.06 (4.42) | 3.44 (2.51) | 1.22 (0.95) | 1.02 (0.93) | 1.02 (0.96) | 1.03 (0.96) |
| 4DCT 10 | 0.82 (0.89) | 3.65 (5.31) | 4.10 (4.12) | 1.01 (1.05) | 0.96 (0.98) | 0.97 (0.99) | 0.99 (0.98) |
| Average | **0.95 (1.15)** | 3.93 (6.07) | 4.63 (5.26) | 1.07 (1.10) | 1.04 (1.12) | *0.98 (1.10)* | 0.99 (1.10) |

Table 7: Ablation study of basis functios number D in KAN-IDIR model on the DIR-Lab dataset. KAN-IDIR model with $D = 28$ delivers best registration quality. Best in **bold**.

| D | 8 | 12 | 16 | 20 | 24 | 28 | 36 |
|---|---|----|----|----|----|----|----|
| Avg. TRE, mm $\downarrow$ | 1.11 | 1.04 | 1.01 | *0.99* | *0.99* | **0.98** | **0.98** |

**Comparison of A-KAN-IDIR vs RandKAN-IDIR.** Table 5 presents a comparative evaluation of A-KAN-IDIR and RandKAN-IDIR across the DIR-Lab, OASIS-1, and ACDC datasets, with metrics including Average TRE, DSC, HD95, and NJD. RandKAN-IDIR outperforms A-KAN-IDIR in most metrics across the datasets, achieving better registration quality and deformation regularity. However, on the ACDC dataset, A-KAN-IDIR achieves a slightly higher Dice score and better HD95. Additionally, A-KAN-IDIR exhibits approximately 20% higher runtime (e.g., 51.2 s vs. 43.1 s on OASIS-1) and VRAM consumption (e.g., 2.64 Gb vs. 2.2 Gb on OASIS-1) compared to RandKAN-IDIR. These results suggest that A-KAN-IDIR's adaptive basis selection mechanism introduces higher computational overhead compared to RandKAN-IDIR's simpler randomized basis sampling, which provides superior registration quality and efficiency for deformable image registration in our experimental setup.

# B   APPENDIX: ADDITIONAL EXPERIMENTAL RESULTS

## B.1   BASELINES IMPLEMENTATION DETAILS AND PARAMETERS SELECTION

For IDIR (Wolterink et al., 2022), ccIDIR (Harten et al., 2023), NODEO (Li et al., 2021), and SINR (Sideri-Lampretsa et al., 2024) we extended the original implementations to all three datasets: OASIS-1, and ACDC for IDIR and ccIDIR; DIR-Lab, OASIS-1, and ACDC for SINR; DIR-Lab and ACDC for NODEO). We ensured that our modifications preserved performance and adhered strictly to the original implementations.

For SINR and NODEO, we identified and fixed several substantial bugs in both codebases that prevented execution on datasets beyond those in the original papers. Pending paper acceptance, we will submit these fixes as pull requests to the respective repositories.

IDIR, ccIDIR, and NODEO were executed using their default parameter configurations without modification. We conducted extensive hyperparameter tuning for these methods on the datasets not used in their original publications; however, this did not yield any substantial performance improvements.

For SINR (Sideri-Lampretsa et al., 2024), we revised the original stopping criterion (NJD > 0.9%) due to observed cases where NJD values initialized high and decreased during optimization. After testing iterations at 300, 500, and 1500, we standardized to 500 iterations as no significant improvements were observed at higher counts. Notably, SINR required additional cropping to landmark min-max coordinates on DIR-Lab to accommodate 48GB GPU memory constraints — the only method necessitating such adaptation.

VoxelMorph (Balakrishnan et al., 2019b) and TransMorph (Chen et al., 2022) were trained without segmentation-based weak supervision. We adjusted their train-test splits to use the last 49 pairs (versus the original 19) to enhance statistical robustness and maintain consistency with other experiments.

For pTV (Vishnevskiy et al., 2016) and CorrMLP (Meng et al., 2024), we directly reproduced the authors' reported values as these represent current state-of-the-art implementations for their respective datasets.

## B.2   ADDITIONAL QUANTITATIVE RESULTS

**DIR-Lab Per-Case Results.** In Table 6 we show the full per-case results for DIR-Lab dataset, following the practice for this dataset in (Wolterink et al., 2022; Harten et al., 2023).

**Ablation for maximum polynomial degree.** In Table 7, we evaluate the performance of the KAN-IDIR $(D)$ model on the DIR-Lab dataset across various values of $D$. The results indicate that a value of $D = 16$ is sufficient for KAN-IDIR to achieve state-of-the-art performance among INR-based methods on the DIR-Lab dataset. For $D = 28$, the performance reaches a plateau, with minimal further improvements observed for higher degrees.

**Ablation for different INR architectures.** To justify the choice of INR in our method, we conducted experiments with several popular INR models, including:

- **ChebyKAN** (Sidharth et al., 2024) — the architecture used in our KAN-IDIR model;
- **Efficient-KAN** — an efficient implementation of classical spline-based KANs (Liu et al., 2024c);
- **FastKAN** (Li, 2024) — a fast KAN variant that replaces third-order B-spline bases with radial basis functions (RBFs);
- **SIREN** (Sitzmann et al., 2020) — the architecture used in IDIR (Wolterink et al., 2022);
- **SL$^2$A** (Heidari et al., 2024) — a novel INR architecture that combines single-layer learnable activation functions with an MLP using standard ReLU activations.

For all models, we tuned the learning rate schedule to achieve optimal performance. All KAN-based models share the same architecture: two hidden layers with 70 neurons each. For SIREN,

Table 8: Evaluation of various INR models on the OASIS-1 dataset. Results for the ChebyKAN architecture correspond to the KAN-IDIR model. The best scores are shown in **bold**, and the second-best in *italic*.

| INR architecture | DSC ↑ | HD95 ↓ | NJD, % ↓ | VRAM, MB ↓ | Runtime, sec ↓ |
|---|---|---|---|---|---|
| ChebyKAN ($D = 12$) | 0.777 (0.17) | 2.08 (0.33) | **0.012 (0.01)** | 1412 | *43.1* |
| ChebyKAN ($D = 28$) | **0.793 (0.16)** | **1.96 (0.32)** | 0.048 (0.03) | 2248 | 63.3 |
| Efficient-KAN (grid $= 24$) | 0.768 (0.18) | 2.16 (0.37) | 0.029 (0.04) | 4134 | 317.0 |
| Efficient-KAN (grid $= 48$) | 0.787 (0.17) | 2.00 (0.33) | 0.198 (0.08) | 7236 | 514.7 |
| FastKAN (grid $= 24$) | 0.768 (0.18) | 2.15 (0.35) | *0.024 (0.03)* | 2322 | 84.4 |
| FastKAN (grid $= 48$) | 0.790 (0.17) | *1.99 (0.34)* | 0.301 (0.12) | 3838 | 146.3 |
| SIREN | 0.784 (0.17) | 2.02 (0.33) | 0.040 (0.03) | **792** | **39.5** |
| SL$^2$A (degree $= 40$) | *0.791 (0.17)* | *1.99 (0.33)* | 0.070 (0.05) | *1016* | 72.1 |
| SL$^2$A (degree $= 72$) | *0.791 (0.17)* | 2.00 (0.33) | 0.732 (0.44) | 1076 | 74.0 |

we adopted the architecture used in IDIR, and for SL$^2$A we retained the original configuration. The number of basis functions ($D$ in ChebyKAN, `degree` in SL$^2$A) and grid points (`grid` in Efficient-KAN and FastKAN) in KAN-based models was selected to approximately match the total parameter count across all networks. To ensure a fair comparison, we used the loss function from the main paper as the training objective for all methods. All models were trained with a batch size of 10,000 points until convergence.

The results of comparing the different INR architectures on the OASIS-1 dataset are shown in Table 8. ChebyKAN – the backbone of our KAN-IDIR method – achieves the best performance in terms of both registration accuracy and runtime. Other INR baselines perform worse than ChebyKAN, which confirms the choice of this architecture for our method. Notably, the two alternative KAN variants demonstrate considerably higher runtimes and VRAM usage than ChebyKAN, highlighting the computational efficiency of Chebyshev polynomials compared to other families of basis functions.

### B.3 ADDITIONAL FIGURES AND VISUALIZATIONS

**Visualization of registration results on brain and cardiac MRI.** Figures 5 and 6 present a qualitative comparison of different methods on the OASIS-1 and ACDC datasets. We extract a 2D slice from the 3D image and visualize the deformation by applying it to a corresponding 2D rectangular grid. The results demonstrate that our method produces smooth deformation fields, yielding images without visual artifacts.

**Visual Appearance of Registration Results for the Suboptimal Seed.** The visual appearance of the registration results that confirms the best performance of the proposed models in terms of seed-dependent stability is demonstrated in Figure 7. The proposed RandKAN-IDIR and KAN-IDIR maintain consistent outlier counts across seeds, with results similar to the optimal seed in Figure 3.

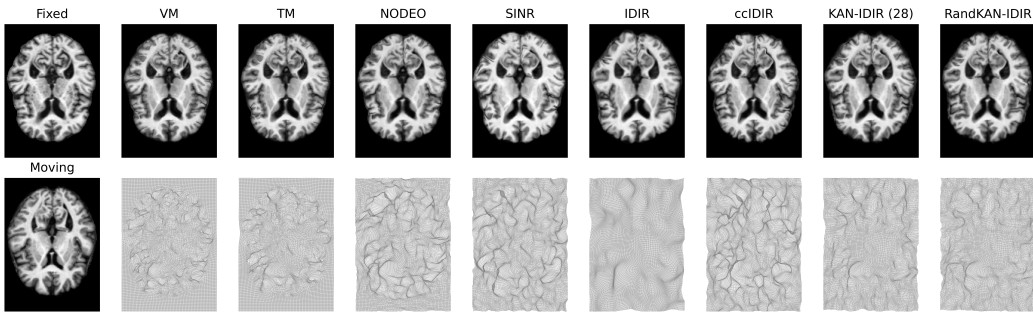

Figure 5: Visual comparison between different image registration models on OASIS-1 dataset. First column consists of fixed image and moving image. Other columns consist (from top to bottom) of warped moving image obtained from model and deformation grid.

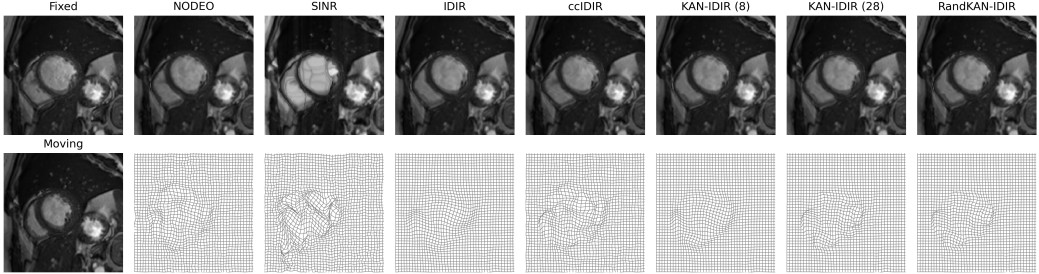

Figure 6: Visual comparison between different image registration models on ACDC dataset. First column consists of fixed image and moving image. Other columns consist (from top to bottom) of warped moving image obtained from model and deformation grid.

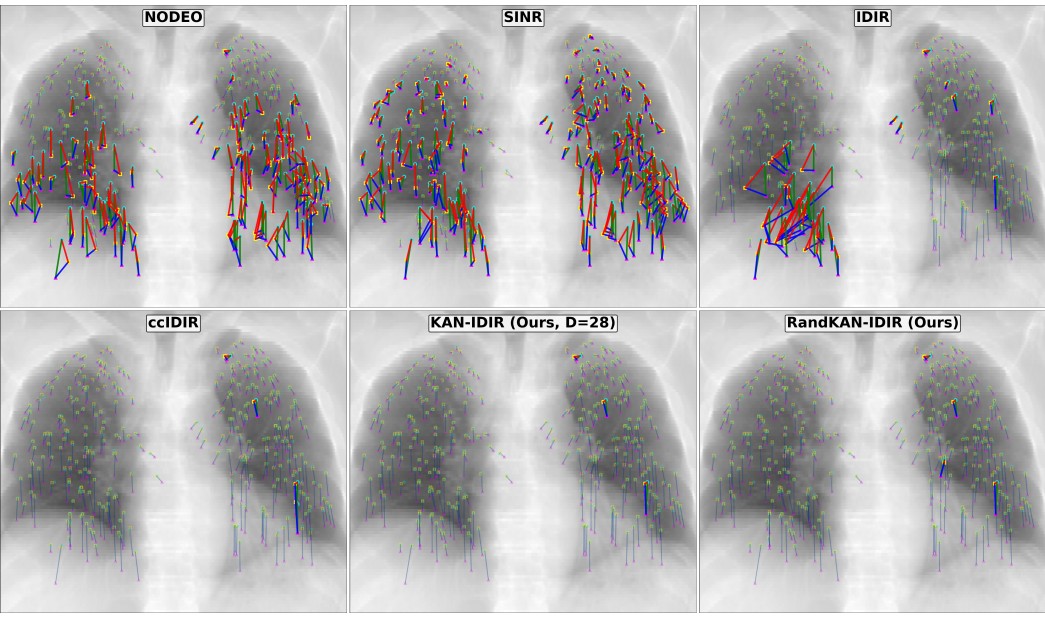

Figure 7: Landmark alignment in DIR-Lab Dataset (Case 7) for fixed, moving, and moved images. Displacements are shown as: blue (method result, moved-to-fixed), green (ground truth, moving-to-fixed), and red (error, moved-to-fixed). Landmarks with errors exceeding 3 mm are highlighted in bold, while others are displayed as transparent. Results reflect each method's suboptimal seed (maximum mean TRE across 10 runs). Proposed KAN-IDIR and RandKAN-IDIR still achieve the fewest outliers, while the other methods demonstrate significant increase.

