# OpenReview forum: "Learning KAN-based Implicit Neural Representations for Deformable Image Registration"
_ICLR.cc/2026/Conference — Submitted to ICLR 2026_

### Official Review · Reviewer_NTqw · 2025-10-28

**Soundness:** 2
**Presentation:** 2
**Contribution:** 2
**Rating:** 2
**Confidence:** 4

**Summary:**

The paper introduces a novel way of representing the transformation of deformable image registration in the context on INRs architectures, using Kolmogorov-Arnold-Networks (KANs). It proposes a new strategy of learning KANs that reduces the the number of learned basis function lowering the overall cost while maintaining competitive performance. The performance is evaluated on a 3 mono-modal datasets and compared against several INR-based methods and 2 cohort-based methods.

**Strengths:**

1. The paper utilizes a new architecture for pairwised INR-based registration that was not used before.
2. It proposed a novel way of learning KAN activations that might be useful out of the context of deformable registration.
3. The performance is evaluated against 3 mono-modal diverse datasets measuring the accuracy, regularity and speed.
4. The language of the paper is clear.

**Weaknesses:**

1. The introduction and related work are rather weak and unfocused failing to motivate the topic effectively. First of all, there is not a clear motivation on what makes KANs a better candidate as a network architecture compared to MLPs in the context of registration. The literature review both in the case of Registration and INRs is not thorough making it difficult to understand what is state-of-the-art in the field.
2. The discussion about iterative optimization methods (classical methods) for registration is somehow missing from the discussion. In my opinion INRs are a sub-category of iterative optimization methods. As a result, I believe they should be mentioned in the introduction and also compared against during the evaluation.
3. There are several inaccuracies in the text, namely:
    1. the paper refers to INR methods as instance-specific optimization methods. However, since registration is happening in pairs I believe that the right term is pairwise optimization methods.
    2. Line 130: *“Second, while segmentation labels enhance performance in brain MRI (Jena et al., 2024), these methods underperform on unlabeled data (e.g., DIR-Lab lung CT (Castillo et al., 2009)), trailing classical (Vishnevskiy et al., 2016) and INR-based (Harten et al., 2023) approaches.”*

        I believe that this is intuitive and that doesn’t diminish their utility. Moreover, I am not sure this is a fair claim since in the case of brains the deep learning methods are trained on segmentation losses and evaluated using dice score (optimization for the metric) while in the case of DIRLab even if they are trained using segmentation labels they are evaluated using TRE on landmarks (the network was not optimized for this metric).

    3. Line 127 - 129: “First, even when trained on large datasets, these image-to-image models remain vulnerable to domain shifts (Jena et al., 2024) - a problem often addressed through even larger datasets (Tian et al., 2024).”

    I am not sure why this is a limitation. If larger datasets are available they can be used to alleviate the domain shift issue mentioned by the authors.
    4. Line 149: *“A fundamentally different (but still instance-specific) approach reformulated registration via neural ODEs in NODEO (Li et al., 2021).”*

    Why do the authors believe that this is a fundamentally different approach than the INRs?
4. I appreciate that the authors experimented with different seeds, but I believe that any learning based method is suffering from the performance discrepancy when choosing different seeds. Is there any reason why this has not been applied to Vmorph and Tmorph?
5. I am not sure I understand the motivation behind the analysis in figure 1 (e.g. What does 54 vs 64 for INR refer to?) This figure and its caption lack clarity.
6. Although there are figures for the lung registration and the landmark errors there are no visualizations of the deformation fields and for the other modalities. Maybe the authors should consider adding them at least in the appendix since visual inspection is a very important way of evaluating whether the final result is accurate.
7. The paper is using a combination of 2 regularization terms:
    1. Where the other methods trained on the same objective? If this is not the case then the evaluation is not very fair and it makes sense why the proposed method is delivering lower folding compared to its counterparts.
8. In the evaluation section I believe that the dice score that is reported is the mean over all structures. It would be usedul to see the mean and std and only demonstrate the seed stability as an ablation over only one application.
9. For completeness I believe that VRAM and speed should also be included in the table for the cardiac data.
10. There is not publicly available repository.

**Minor points**

1. Is there any drawback due to the choice of Chebychev polynomials instead of the original splines?
2. Is there any intuition why the different KAN strategies behave differently?
3. It is not clear whether the INR methods are fitted till convergence or over fixed steps and whether the learning rate and omega of siren was tuned to the application. Finally, the choice of the batch size affects both the memory requirements, the convergence and the performance. It would be nice to know how all these hyperparameters were chosen to understand whether the evaluation is fair.

**Questions:**

Please see the weaknesses.

**Preliminary Justification:**

The paper presents a conceptually interesting approach to a new way of representing the transformation of registration based on KANs.

However, I believe that the paper quality is insufficient to be accepted in ICLR this year. I believe the authors have to put substantial effort into enhancing the motivation, the literature review, the storyline, experiments, and discussion, as well as to include more baselines (classical methods) and qualitative results.

Given these shortcomings, I recommend rejection in its current form. Nonetheless, the underlying idea is promising, and I encourage the authors to strengthen the paper and resubmit it to a future venue. Finally, given the structure of the proposed work, I believe that this would be a better fit in a computer vision conference.

---

> ### Author Response · Authors · 2025-11-25
>
> We thank the reviewer for the detailed review and for highlighting both strengths and needed improvements. We address each point below. Note, that we highly appreciate every single comment and consider them all valuable. We purposly omit the "thanks" in every comment response to make our responses more concise. We hope that it won't sound less polite.
>
> **W1. (motivation of KANs vs MLPs; intro/related work not thorough)**
>
> We have modified the introduction and the related works sections and hope that now it better moticates the choice of KANs as an architecture for DIR task. Besides that, we added more references to recent INR works and classical iterative DIR approaches and hope that the text now better reflects the SOTA in the field.
>
> **W2. (“classical iterative methods missing” and INRs as a sub-category of iterative optimization)**
>
> We have added the review of the classical iterative methods to the literature review section in the revised version of the manuscript.
>
> **W3.1. (“instance-specific” vs “pairwise optimization”)**
>
> We agree. The text is revised accordingly.
>
> **W3.2. (comparison between segmentation-supervised brain MRI and TRE on DIR-Lab)**
>
> Our intention was not to imply that segmentation-supervised DL methods lack utility, but rather to highlight that such methods tend to underperform on unlabeled datasets such as DIR-Lab lung CT. (Jena et al. ,2024) note that DLIR methods often overfit to the specific label structures present in the training data. When trained without additional supervision, these methods can perform worse than classical iterative approaches in certain domains. For example, (Tian et al., 2023, Table 2) report a comparison of widely used iterative and learning-based methods on the DIR-Lab dataset: the best DLIR method achieves a TRE of 1.26 mm, whereas the state-of-the-art iterative method, pTV, achieves a 30% lower error. Importantly, for DIR-Lab, both classical and learning-based methods are trained without extra supervision such as segmentations or keypoints; that is, neither is explicitly optimized for the evaluation metric during training. This makes the evaluation protocol fully fair in this case.
>
> References:
>
> >Tian, L., Greer, H., Vialard, F. X., Kwitt, R., Estépar, R. S. J., Rushmore, R. J., ... & Niethammer, M. (2023). GradICON: Approximate diffeomorphisms via gradient inverse consistency. In Proceedings of the IEEE/CVF Conference on Computer Vision and Pattern Recognition (pp. 18084-18094).
>
> **W3.3. (domain shift remark; “why is this a limitation if larger datasets exist?”)**
>
> As we wrote, the authors of (Jena et al., 2024) have shown that domain shift problem cannot be fully solved using dataset-specific methods even for complex architectures. Moreover, for some domains large datasets are not available at all (e.g. DIRLAB is small), which is also stated in our sentence.
>
> **W3.4. (“why call NODEO fundamentally different?”)**
>
> We agree, NODEO indeed is not fundamentally different, we rephrased the sentence accordingly.
>
> **W4. (seed-sensitivity: “why not also run VoxelMorph/TransMorph with multiple seeds?”)**
>
> We experimented with different seeds for VoxelMorph amd TransMorph and did not observe large discrepancy for these models. Thus we did not include these experiments into the paper.
>
> **W5. (Figure 1 clarity; “what do 54 vs 64 refer to?”)**
>
> Figure 1 is meant to assess robustness to random seed initialization, which can noticeably affect optimization-based registration. The numbers “54 vs. 64” refer to the count of landmarks with TRE > 3 mm under each method’s best-performing seed. We now make this explicit in the caption. We also clarify the seed-to-seed variability: KAN-IDIR increases from 54 to 58 outliers between its best and worst seeds (difference of 4), while IDIR increases from 64 to 120 (difference of 56). This illustrates that KAN-IDIR is markedly more stable to initialization. The caption has been updated accordingly.
>
> **The rest of the weaknesses and minors are responded in the next comment below.**

---

> ### Author Response · Authors · 2025-11-25
> **Continuation of the response to reviewer NTqw**
>
> **W6. (missing deformation-field visualizations for other modalities)**
>
> We've added the visualizations for the OASIS and ACDC datasets to the appendix (Sec. B.3, figures 5 and 6) in the revised version of the text.
>
> **W7.1. (regularization fairness: “were other methods trained with the same objective?”)**
>
> We believe that in image registration the configuration of loss function and choice of regularization terms are the important parts of the methodology introduced by method authors. We also believe that authors of corresponding papers performed all necessary ablations and the choice of regularizers is well motivated. Thus we tried to use the original methods in all details for comparison. However, we appreciate the reviewer's comment and added another ablation study into the appendix (Sec.B.2, Table 8), where we used our regularizer along with different INRs acrhitectures.
>
> **W8. (Dice reporting: mean/std; limiting seed-stability analysis to one application)**
>
> Our reported Dice standard deviation is computed across all anatomical structures and all random seeds. We chose this aggregated measure because it naturally captures both seed stability and structural variability, and in our setting it does not obscure either aspect.
>
> **W9. (VRAM + speed missing for cardiac data)**
>
> We agree with this claim. In the revised version of the paper, we have included VRAM consumption and runtimes for the ACDC dataset.
>
> **W10. (no public repository)**
>
> We have included the complete codebase into the supplementary materials so the reviewers could get familiar with it. And there is a public repository with our code. However, it's forbidden to include it to the main text of ICLR paper as it breaks anonymity.
>
> **M1. (possible drawbacks of Chebyshev vs splines)**
>
> We provide a comparison of KANs with Chebyshev polynomials versus KANs with splines in the ablation study for different INR architectures, which we have added to the appendix (Sec.B.2, Table 8). Currently, we observe no drawbacks to using Chebyshev polynomials instead of B-splines, as they offer higher performance and greater computational efficiency.
>
> **M2. (intuition for differences between KAN strategies)**
>
> We believe that our randomized strategy for basis selection performs well because most natural signals are inherently sparse, enabling effective performance of sparse approximation and compressed sensing algorithms. The strong performance of the random basis selection arises because the reduced number of basis functions is compensated by higher polynomial degrees, which capture high-frequency information more effectively.
>
> As for adaptive learning of basis function indices, in our experiments it performed worse due to unstable training process. We add noise to the learnable logits to avoid local minima at the start of training, but this noise introduces instability. Nevertheless, in theory, more effective algorithms for adaptive learning could be developed, which remains a topic for future research.
>
> **M3. (clarification on convergence vs fixed steps; tuning of SIREN hyperparams; batch size choice)**
>
> For the SIREN model used in the IDIR method, the learning rate and value of the $\omega$ parameter were tuned by the original authors, and we use these default values in our experiments. For KAN-based models, we experimented with different numbers of optimization steps and selected 1500 as the optimal value, as the performance gains beyond this point were negligible. For the batch size, a value of 10,000 is commonly used across papers on image registration with INRs (e.g., IDIR, ccIDIR). We therefore adopt the same value in our method as well.
>
> Thank you again for the constructive comments. We believe the planned clarifications, additional experiments, and improved figures will address these concerns and substantially strengthen the paper.

---

> ### Author Response · Authors · 2025-11-27
>
> Dear Reviewer NTqw,
>
> Thank you for your insightful review and helpful suggestions. We’ve provided a set of detailed responses to all your comments and would appreciate it if you could review them when time permits. Should anything require additional clarification, we’d be happy to address it.
>
> Sincerely,
>
> Authors of paper 14113

---

### Official Review · Reviewer_NKPY · 2025-10-28

**Soundness:** 2
**Presentation:** 1
**Contribution:** 2
**Rating:** 2
**Confidence:** 4

**Summary:**

This paper introduces a KAN-based Implicit Neural Representation (INR) framework for deformable image registration and compares its performance to SIREN-based INRs. Moreover, the authors ablate architectural variants of the KAN network with respect to its (learnable) basis function.

**Strengths:**

- Using KANs over traditional, MLP-based architectures for INRs is a refreshing and novel perspective for deformable image registration.
- Based on the authors' presentation, their proposed method outperforms all baseline models for all datasets.

**Weaknesses:**

### Presentation of the method within the context of other works (Introduction/Related Work):

The structure of the introduction and related work does not effectively motivate the usage of KANs in the context of deformable image registration, and it lacks the placement of the proposed framework within this domain.

To be precise:
* The authors provide a lengthy introduction to state-of-the-art cohort-based registration frameworks, such as VoxelMorph, yet they focus on iterative settings. They should instead focus on placing their proposed method within this context.
* The motivation for choosing KAN over an MLP in the context of deformable registration is not clearly stated—what specific features do KANs offer that MLPs do not?
* While seminal works like [1] have employed SIREN, a plethora of other INR architectures have been proposed since—which may be equally relevant (to KANs) for this task—such as those utilizing Gaussian activation functions, FINER, BACON, MFN, etc. [2-6]. Since the primary contribution of this paper is to use KANs instead of MLP-based INRs, the authors should discuss (and later ablate) these architectures in this context, and not focus only on SIREN. For instance, [2] remains of particular relevance since they group ReLU activations to be a spline, which may be adjacent to this work, even though the authors do not use spline-based KANs.

Moreover, the KAN literature has also grown significantly over the past year. Since one of the authors' main contributions is the introduction of their RandKAN model, it would be beneficial to explain the elements of different KAN variants with more rigor and also contrast their performance in experiments. For instance, why did the authors choose the Chebyshev KAN architecture over the traditional KAN (Liu et al.) using B-splines, given that methods like FFD [7] typically use B-Splines? Is there any intuition for this? While the authors state that this increases the speed, did they ablate this decision?

### Experimental Setup:

* The authors use entirely different regularization than methods like IDIR [1]. The motivation for using other regularization (e.g., no bending energy) remains unjustified. A fair experimental setup would integrate all architectures into the same framework and evaluate all models for the same model footprint on the problem, requiring hyperparameter tuning for all architectures within the same framework. For instance, the KAN models are much smaller, which may cause them to overfit much less to noise and thus exhibit much less folding. How does folding behave once the model size is increased?
* Missing classical, iterative registration pipelines for DIRLAB experiments.
* The authors report that default parameters were used for, e.g., IDIR for all datasets, resulting in suboptimal performance since IDIR's hyperparameters were optimized for lung registration, and not for brain (OASIS) or cardiac datasets.
* No qualitative results are provided for the OASIS and ACDC datasets.

Given these substantial weaknesses, I recommend a thorough revision of the Introduction/Related Work and the experimental setup for this manuscript.

[1] Wolterink JM, Zwienenberg JC, Brune C. Implicit neural representations for deformable image registration. MIDL'22

[2] Liu Z, Zhu H, Zhang Q, Fu J, Deng W, Ma Z, Guo Y, Cao X. Finer: Flexible spectral-bias tuning in implicit neural representation by variable-periodic activation functions. CVPR'24

[3] Fathony R, Sahu AK, Willmott D, Kolter JZ. Multiplicative filter networks. ICLR2020

[4] Lindell DB, Van Veen D, Park JJ, Wetzstein G. Bacon: Band-limited coordinate networks for multiscale scene representation. CVPR22

[5] Ramasinghe S, Lucey S. Beyond periodicity: Towards a unifying framework for activations in coordinate-mlps. ECCV'22

[6] Shenouda J, Zhou Y, Nowak RD. ReLUs are sufficient for learning implicit neural representations. ICML'24

[7] Rueckert D, Sonoda LI, Hayes C, Hill DL, Leach MO, Hawkes DJ. Nonrigid registration using free-form deformations: application to breast MR images. TMI'02

**Questions:**

1.  How is sparse approximation related to pruning literature, both within and outside the scope of KANs and neural networks?
2.  How do spline-based KANs compare to the Chebyshev KANs in terms of performance (excluding speed)?

---

> ### Author Response · Authors · 2025-11-25
>
> Thank you for the thoughtful review and for highlighting both strengths and the areas needing improvement. We respond to each point below. Note, that we highly appreciate every single comment and consider them all valuable. We purposly omit the "thanks" in every comment response to make our responses more concise. We hope that it won't sound less polite.
>
> ## W1. (intro/related work structure; insufficient motivation for KANs; unclear placement within registration literature)
>
> We agree that the introduction and related work should more clearly motivate the use of KANs for deformable image registration and position our proposed framework within this domain. We have modified the introduction and the related works sections in the revised version of the manuscript.
>
> **W1.a. (“focus on cohort-based methods like VoxelMorph is too long; should instead position our iterative INR approach clearly”)**
>
> We have added more references to recent INR works and classical iterative DIR approaches and hope that the text now better reflects the SOTA in the field.
>
> **W1.b. (“motivation for choosing KAN over MLPs not clearly stated”)**
>
> We addressed this point in the Section 2 of our paper:
> > "Several studies <...> have demonstrated that KANs outperform MLPs in accurately approximating functions of a physical or mathematical nature, like special functions or solutions to partial differential equations. Image registration can be formulated as finding the solution an Euler-Lagrange system of a specific target functional <...>, providing strong justification for using KANs in this context."
>
> We hope that revised version better motivates the choice of KANs as an architecture for DIR task.
>
> **W1.c. (“only SIREN discussed; missing discussion of other INR variants such as FINER, MFN, BACON, Gaussian activations, ReLU-as-splines, etc.”)**
>
> We have added the review of the mentioned (and not only) INRs in the related work. We also adapted some of the mentioned INRs to DIR task and included the experimental results with to appendix (Sec. B.2, Table 8).
> At the same time, we would like to clarify that in INR-based approaches the activation function is a core part of the methodology. As a result, replacing the INR effectively defines a new DIR method. A comprehensive evaluation of all possible INR architectures across multiple datasets is therefore beyond the scope of the present paper. Nonetheless, we appreciate the reviewer’s point and have incorporated additional comparisons.
>
> **W1.d. (“KAN literature has grown; need deeper explanation of different variants and why Chebyshev KAN instead of spline-based KAN; any ablation?”)**
>
> We have adapted other KAN variants and added the corresponding experimental results to the appendix (Sec. B.2, Table 8) to illustrate how these architectures behave compared to the proposed one.
>
> **The rest of the weaknesses and questions are responded in the next comment below.**

---

> ### Author Response · Authors · 2025-11-25
> **Continuation of the response to the reviewer NKPY**
>
> ## W2. (experimental setup concerns: different regularization; fairness; model footprint; potential impact on folding)
>
> **W2.a. (“why we changed regularization used in IDIR; need justification”)**
>
> Total variation regularization is more computationally efficient than bending energy, as it requires fewer backward passes through the network to compute the regularizer analytically. This is reflected in the runtime comparison: IDIR (Table 3) runs nearly six times slower than SIREN with our loss (Table 8) – 260.8 s vs. 39.5 s. This substantial reduction in runtime is the primary motivation for our choice of regularization. Moreover, bending energy may oversmooth deformation fields, leading to degraded registration quality. For example, IDIR achieves a DSC of 0.762 on OASIS (Table 3), whereas SIREN with our regularizer reaches 0.784 (Table 8), representing an improvement of about 3%. A visual example of the oversmoothed deformation produced by IDIR is shown in the Appendix (Figure 5).
>
> **W2.b. (“all architectures should ideally share same framework + hyperparameter tuning; smaller KANs may fold less due to capacity, what happens if model size increases?”)**
>
> We believe that in image registration the configuration of the loss function and choice of regularization terms are the important parts of the methodology introduced by method authors. We also believe that authors of corresponding papers performed all necessary ablations and the choice of regularizers and hyperparameters are well motivated. Thus we tried to use the original methods in all details for comparison. However, we appreciate the reviewer's comment and added another ablation study into the appendix (Sec. B.2, Table 8) where we used our regularizer along with different INRs, including KANs.
>
> As for the KANs size, we chose the architecture of our KANs to have nearly the same number of parameters as in the baseline models.  We included this clarification into the updated version of the text as well.
>
> **W2.c. (missing classical iterative baselines for DIR-Lab)**
>
> As stated in the paper, we chose the best performing method for our comparisons which is pTV by Vishnevskiy et al. The complete comparison of pTV with the other classical approaches is given in that paper. We did not copy those results to our work due to space limitations.
>
> **W2.d. (“default IDIR parameters used outside lung CT → suboptimal; should re-tune for OASIS/ACDC”)**
>
> We thoroughly experimented with IDIR parameters on all datasets. However, parameter tuning did not give any substantial increase in metrics for OASIS and ACDC.
>
> **W2.e. (no qualitative results for OASIS or ACDC)**
>
> We agree. We have added the visualizations for OASIS and ACDC to appendix (Sec. B.3, figures 5 and 6).
>
> **Q1. (relation between sparse basis selection and pruning;)**
>
> Sparse approximation provides the mathematical basis for most pruning methods. In classical signal processing and compressed sensing, one seeks a representation for a signal using as few non-zero coefficients as possible in a chosen dictionary. We believe that neural network pruning uses the same idea: starting from a full model with many parameters, it searches for a sparse subnetwork that approximates the original function with minimal loss, via magnitude pruning or sparse training.
>
> In this sense, pruning in standard MLP/CNNs and pruning/sparsification in KANs are just different realizations of sparse approximation: both perform basis/parameter selection under an accuracy-sparsity trade-off. In our setting, adaptive selection of polynomial bases in KAN-INRs for deformable registration can be viewed as choosing a sparse set of basis functions that is enough to represent the deformation field with satisfactory approximation error. We think that it conceptually aligns our method with broader sparse approximation and pruning literature.
>
> **Q2. (comparison of spline-KAN vs Chebyshev-KAN beyond speed)**
>
> We provide a comparison of Chebyshev KANs versus KANs with splines in the ablation study, which we have added to the appendix (Sec. B.2, Table 8). Currently, we observe no drawbacks to using Chebyshev polynomials instead of B-splines, as they offer higher performance and greater computational efficiency.
>
> Thank you again for these constructive points. We believe that the additional clarifications, expanded related work, and improved experimental setup directly address the concerns you raised and will significantly strengthen the paper. Let us know if a particular point needs further elaboration.

---

> ### Author Response · Authors · 2025-11-27
>
> Dear Reviewer NKPY,
>
> We appreciate your valuable comments on our manuscript. We have now posted detailed replies addressing each point and would welcome your feedback if you have a moment to look them over. Please feel free to let us know if any issues remain unclear.
>
> Sincerely,
>
> Authors of paper 14113

---

### Official Review · Reviewer_bzhm · 2025-10-29

**Soundness:** 3
**Presentation:** 3
**Contribution:** 2
**Rating:** 4
**Confidence:** 5

**Summary:**

This paper proposes KAN-IDIR and RandKAN-IDIR, the first integration of Kolmogorov–Arnold Networks (KANs) into implicit neural representation (INR) based deformable image registration.

**Strengths:**

Comprehensive evaluation: Evaluated on three modalities/organ types with multiple metrics (TRE/DSC/HD95/NJD), per-case tables, seed variability, and ablations on basis size and loss weights.

Seed-stability analysis: The paper explicitly studies seed-dependent variability and demonstrates improved robustness, an important but often neglected INR property.

**Weaknesses:**

1. In figure 1, the visual differences between the first two panels and the last two panels in Figure 1 are not explained; it is difficult to understand what each subfigure is intended to illustrate.
2. The contribution reads largely as “apply Chebyshev KAN architecture to INR for registration” with no substantial methodological modifications targeted specifically to registration (beyond RandKAN sampling). This reduces perceived novelty.
3. In figure 3, the regularization loss should be imposed to the learned displacement, not on the addition of the displacement and the input coordinate.
4. In Table 2, the results of pTV method are not reported.
5. The claim of “best or best-in-class across all datasets” is overstated: classical pTV remains best on DIR-Lab TRE in Table 1, and dataset-trained methods outperform on some metrics (OASIS).
6. It is unclear whether reported runtimes are end-to-end (training per instance) or inference only, and how they scale with image resolution.
7. The paper adopts Chebyshev-based KANs but provides limited ablation comparing different KAN variants or other INR bases (e.g., SIREN, Fourier features, B-spline KAN). It is therefore unclear whether Chebyshev KAN is the best or merely sufficient.

**Questions:**

How robust is the method to multimodal intensity differences (e.g., CT→MR registration)? Have you tried cross-modality experiments?

---

> ### Author Response · Authors · 2025-11-25
>
> Thank you for the constructive review and for the positive comments on evaluation depth and seed-stability analysis. We address each weakness below. Note, that we highly appreciate every single comment and consider them all valuable. We purposely omit the "thanks" in every comment response to make our responses more concise. We hope that it won't sound less polite.
>
> **W1. (Figure 1 is unclear; differences between the first two and last two panels not explained)**
>
> We agree. We now state clearly in the caption that the first two panels correspond to each method’s best seed (lowest mean TRE) and the last two panels correspond to the worst seed (highest mean TRE). This layout is intended to show how each method behaves under favorable versus unfavorable optimization conditions. We also clarify the observed differences: IDIR varies widely across seeds (64 to 120 outliers), while KAN-IDIR remains stable (54 to 58). The caption has been updated accordingly.
>
> **W2. (“contribution feels like simply applying Chebyshev KANs to registration; limited methodological novelty beyond RandKAN”)**
>
> We believe that our contribution is novel and hope that it will be useful for the community. We first introduced KANs into INRs for registration tasks, making the SOTA results within the INR methods and for some datasets overall SOTA results. Second, there are not many works on KANs in INRs in general. And finally, besides RandKAN we comprehensively researched adaptive choice of bases in KANs to optimize those models (see A-KAN section in appendix), and RandKAN is what we ended up, after a substantial research in this direction.
>
> **W3. (in Figure 3, regularization should be applied to the learned displacement, not displacement + coordinates)**
>
> We believe this is not an error, as the regularization may also depend on the deformation field (e.g., when using a Jacobian determinant loss). Moreover, the Jacobians of the displacement field and the deformation field differ only by the identity matrix, so the regularization term can always be written in terms of the deformation field jacobian, which is how we present it in our figure for clarity.
>
> **W4. (Table 2 missing pTV results)**
>
> The pTV approach is tailored for lung data, especially solving the sliding motion problem. There is no publically available adaptation of this method for OASIS dataset. Moreover, we tried to compare our approaches to the current SOTA or widely-used approaches for each dataset, while pTV is not reported to perform well on OASIS.
>
> **W5. (claim of “best or best-in-class across all datasets” is overstated; pTV still best on DIR-Lab; dataset-trained methods outperform on some OASIS metrics)**
>
> We agree. pTV is the best for DIR-lab outperforming any existing DL-based methods, both dataset-trained or pairwise. However, several registration INRs appeared and were evaluated and reported for this dataset. And our method is best among all INRs (+ pariwise NODEO). That's why we report it as "best-in-class".
>
> **W6. (unclear whether runtimes measure full per-pair optimization or inference; unclear scaling with resolution)**
>
> For INR-based methods (including NODEO), where a separate model is trained for each image pair, there is no train/inference split. Therefore, we report the end-to-end runtime, including the model optimization time for the specified number of iterations. This follows common practice in the INR literature. For dataset-trained methods such as VoxelMorph and TransMorph, we report training and inference times separately.
>
> As for the image size, the INRs (except NODEO and SINR) are resolution-robust by design which is considered to be one of the nice properties of this family of methods. The computational costs of NODEO and SINR scale with image resolution in both memory and runtime. Similarly, for learning-based methods such as VoxelMorph and TransMorph, both training and inference costs increase with image resolution.
>
> **W7. (limited ablation across different KAN variants / INR bases; unclear whether Chebyshev KAN is optimal rather than sufficient)**
>
> We have adapted other KAN variants (within what was feasible during the rebuttal period) and added the corresponding experimental results to the appendix (Section B.2) to illustrate how these architectures behave compared to the proposed one.
>
> **Q1. (robustness to multimodal intensity differences; CT→MR or other cross-modality scenarios)**
>
> We believe that INRs are possible to extend to multi-modal registration scenarios, but we did not experiment with that yet. However, that is our plan for future work, which is out of scope for this paper.
>
>
>
>
>
>
> Thank you again for the helpful feedback. The clarifications and additions we plan to include directly address these points and we believe they will significantly strengthen the paper.

---

> ### Author Response · Authors · 2025-11-27
>
> Dear Reviewer bzhm,
>
> Thank you once again for your thoughtful and constructive feedback. We’ve uploaded our detailed responses to your comments and would be grateful if you could review them at your convenience. Please let us know if any points would benefit from further clarification.
>
> Sincerely,
>
> Authors of paper 14113

---

### Official Review · Reviewer_Mfq6 · 2025-11-01

**Soundness:** 3
**Presentation:** 3
**Contribution:** 2
**Rating:** 6
**Confidence:** 4

**Summary:**

The authors consider the task of deformable image registration using the well known method INRs to model the deformation/displacement field.

Contrary to previous work they use an other network architecture for the INR, in this case a Chebyshev KAN. They explored the use of fixed, randomized and adaptive selection of polynomial basis functions for their network. They evaluate their task on three datasets, including 4D lung CTs, brain MRIs and cardiac MRIs. They show that for the selected datasets their method slightly outperforms previous work with other network architectures, while consuming a little bit less of computational resources.

**Strengths:**

The paper is well written in all sections and their architecture is clearly explained The authors report a performance improvement with a lower resource consumption. They shared their implementation and used publicly available datasets which is a plus for the reproducibility, and they promise to make their code publicly available. In the analysis the authors provide multiple ablations across various settings and parameter ranges and also show some qualitative examples of their method compared to others.

**Weaknesses:**

The authors only investigate one type of networks architectures (the Chebyshev KANs) and compare them against other existing types from other publications. Since this was the only focus of this paper, it would have been nice to see some broader comparison across other network architectures that have been proposed for INRs. One of the previous work they base their work on (Wolternik, 2022) used SIREN, but since, there have many other architectures have been published (WIRE, FINER etc) that easily could have been explored in the same work, so the motivation for only focusing this particular architecture remains unclear.

**Questions:**

- The authors should compare also compare their network with other architectures like WIRE, FINER,... How do they compare to their chosen approach? Justify your choice!

---

> ### Author Response · Authors · 2025-11-25
>
> Thank you for the positive evaluation of the writing quality, clarity of the architecture, ablations, and reproducibility aspects. We address the main weakness and the corresponding question below.
>
> We have added the discussion of other INRs to literature review of the revised manuscript. In addition, we have added another ablation study into the appendix (Sec.B.2, Table 8) where we adapted other KANs and other INRs to DIR task (which was feasible wihin the short scope of rebuttal period). We hope that it substantially improve both the motivation of the proposed architecture and experimental results of the work.
>
> However, we believe that in INR-based approaches the activation function is a core part of the methodology. As a result, replacing the INR effectively defines a new DIR method, as also noted in prior work. A comprehensive evaluation of all possible INR architectures across multiple datasets is therefore beyond the scope of the present paper and would constitute a substantially larger study.
>
> Thank you again for the helpful feedback.

---

> ### Author Response · Authors · 2025-11-27
>
> Dear Reviewer Mfq6,
>
> Thank you again for your helpful feedback. We’ve posted detailed responses to your comments and would appreciate it if you could take a quick look when you have a moment. Please let us know if anything needs further clarification.
>
> Sincerely,
>
> Authors of paper 14113

---

### Author Response · Authors · 2025-11-30
**Official summary comment for the Area Chair**

Dear Area Chair,

Over the discussion period we worked intensively to address every concern raised by the reviewers. This required substantial additional experimentation, reimplementation of several baselines, new visualizations, and multiple writing improvements. Overall, in 1 week we reimplemented and adapted 4 new baselines, added 2 new figures and 1 table, resulting in almost 1 additional page in the main text and 2 additional pages in appendix (the differences are highlighted in the revised PDF). We believe that we addressed **all** concerns raised by the reviewers in the point-by-point answers below, but we'll be happy to elaborate further if anything still needs clarification.

Below we provide a short summary of the revisions, which we hope will assist in the final evaluation.

### Methodology
We addressed all methodological concerns raised by the reviewers. In particular:

- We clarified the experimental setup and resolved all related concerns raised by reviewers **NKPY** and **NTqw**.
- We expanded the intuition behind our approach and clarified its relationship to sparse approximation, as requested by reviewers **NKPY** and **NTqw**.
- We addressed all remaining methodology related questions from reviewers **bzhm**, **NKPY**, and **NTqw**.

### Writing
We improved the Introduction and Related Work sections in response to comments from reviewers **NTqw** and **NKPY**. Following **NTqw**, we added missing discussion of classical iterative methods, corrected terminology, and clarified claims related to evaluation protocols and domain shift in DLIR methods. Following **NKPY**, we expanded the INR literature to include additional architectures and provided clearer explanations of KAN variants. We also clarified our architectural decisions and improved the clarity and accuracy of the text throughout.

### Additional experiments
The main concern from reviewers about experiments was the absence of comparisons with other INR architectures. We addressed this in the Appendix (Sec. B.2, Table 8), where we compared our models under the same loss and regularizer. We added two KAN-based baselines (EfficientKAN and FastKAN) and two MLP or hybrid models (SIREN and SL²A), alongside our Chebyshev KAN, to justify its use as the backbone of our method.

### Figures
To address reviewers’ comments, we added two new figures and corrected one existing figure. Reviewers **bzhm** and **NTqw** noted that Figure 1 lacked clarity, so we completely revised its caption and added a y axis title. Reviewers **NKPY** and **NTqw** requested additional qualitative results for OASIS and ACDC, which we added in Appendix Sec. B.3 (Figures 5 and 6).

Thank you very much for your time and consideration.

---

### Meta-Review · Area_Chair_sp5v · 2025-12-17

**Summary:**

Reviewers raised concerns that the methodological contribution is incremental, largely centered on swapping the INR backbone.

Several reviewers also questioned the fairness and clarity of the experimental setup, the strength of motivation and positioning relative to prior INR and classical registration methods, and the absence of broader architectural and classical baseline comparisons.

While the rebuttal addressed many presentation and experimental gaps, these concerns informed a cautious overall assessment.

**Reviewer Concerns:**

The rebuttal addressed several major concerns by adding broader comparisons across INR and KAN variants, improving the introduction and related work to better motivate KANs for registration, clarifying terminology and runtime reporting, fixing unclear figures, and adding missing qualitative results for OASIS and ACDC. The authors also tempered overstated claims and expanded experimental details, including additional baselines and ablations, which substantially improved clarity and completeness.

However, some concerns remain only partially resolved. In particular, reviewers still questioned whether the experimental protocol fully isolates the effect of the KAN architecture versus differences in regularization and objectives, and whether the contribution goes beyond adopting a new INR backbone with limited task-specific innovation. Cross-modality robustness was not evaluated and remains future work, and comparisons to classical iterative registration methods beyond the lung dataset remain limited.

**Reviewer Scores:**

Reviewers are not excited about this paper in general, and have not participated in the discussion as well.

---

### Decision · Program_Chairs · 2026-01-26

Reject